# Hemodynamic Markers Predict Outcomes a Decade After Acute Coronary Syndrome

**DOI:** 10.3390/jcm14186627

**Published:** 2025-09-19

**Authors:** Andrzej Minczykowski, Oskar Wojciech Wiśniewski, Tomasz Krauze, Adam Szczepanik, Agnieszka Banaszak, Przemysław Guzik, Andrzej Wykrętowicz

**Affiliations:** 1Department of Cardiology—Intensive Therapy, Poznan University of Medical Sciences, 60-355 Poznan, Poland; anmin@ump.edu.pl (A.M.); tomaszkrauze@ump.edu.pl (T.K.); aslab@op.pl (A.S.); aban3@op.pl (A.B.); pguzik@ptkardio.pl (P.G.); awykreto@ptkardio.pl (A.W.); 2Doctoral School, Poznan University of Medical Sciences, 60-812 Poznan, Poland

**Keywords:** acute coronary syndrome, arterial stiffness, echocardiography, hemodynamic markers, long-term prognosis, myocardial infarction, ventricular–arterial coupling

## Abstract

**Background:** Previous research from our group demonstrated that novel hemodynamic indices can predict 3–5-year mortality risk in myocardial infarction survivors. Building on these findings, we assessed the long-term prognostic value of these markers over a 10-year follow-up period. **Methods**: We conducted a prospective study involving 569 consecutive acute coronary syndrome (ACS) patients admitted within 12 h of symptom onset, all presenting with >50% coronary artery stenosis. Hemodynamic indices were assessed using echocardiography to measure ejection fraction (EF), global longitudinal peak systolic strain (GLPSS), and ventricular–arterial coupling (VA coupling). Excess aortic pressure (excess_PTI_) was evaluated via radial tonometry, while local arterial stiffness was assessed by pulse wave velocity (PWV) through carotid ultrasonography. The primary outcome was all-cause mortality over a 10-year follow-up period. **Results**: Over a median follow-up of 3249 days, 172 patients reached the primary endpoint (death). Deceased individuals were older and exhibited lower EF, impaired VA coupling, higher excess_PTI_, and a lower PWV/GLPSS index compared to survivors. In multivariate Cox proportional hazards analysis, EF, VA coupling, excess_PTI_, and PWV/GLPSS index were independently associated with all-cause mortality over a 10-year follow-up period. **Conclusions**: This study highlights the significant long-term prognostic value of novel hemodynamic indices, including VA coupling, PWV/GLPSS index, and excess_PTI_, in predicting 10-year all-cause mortality in ACS patients.

## 1. Introduction

Despite significant advancements in the treatment of acute coronary syndromes (ACS), patients remain at substantial risk for cardiovascular complications, including death. Time is an essential uncontrolled factor and appears to be associated with a rise in mortality as time increases since the index event [1]. Long-term follow-up validates the utility of various prognostic factors in post-ACS patients.

Modern interventional and pharmacological treatment of ACS helps to preserve left ventricular (LV) function and improve overall prognosis. It is now recognized that the majority of ACS survivors represent a relatively low-risk population compared with historical cohorts due to contemporary treatment. Echocardiography remains a vital imaging modality in the evaluation of patients with ACS. In recent years, both traditional markers, such as ejection fraction (EF), and novel descriptors of LV function, such as global longitudinal peak systolic strain (GLPSS), arterial stiffness–pulse wave velocity (PWV)/GLPSS index, ventricular–arterial coupling (VA coupling), or excess pressure-time integral (excess_PTI_), have been used as potential predictors of future cardiovascular complications [2,3,4,5,6]. All mentioned parameters are measured non-invasively, which makes them safe, easily-accessible, and cost-effective. Precise measurement techniques are presented in the Material and Method section.

VA coupling describes the continuous interplay between LV and arterial tree, thus expressing global cardiovascular efficiency [7]. It is quantified as a VA coupling ratio, depending directly on effective arterial elastance (Ea; representation of arterial load) and inversely on LV end-systolic elastance (Ees; a measure of LV contractility). The VA coupling ratio that equals/nearly equals 1.0 is considered optimal for stroke work generation [8,9]. Disproportionately high Ea to Ees ratio is often observed in patients after myocardial infarction (MI), indicating VA decoupling, and independently associated with adverse cardiac events and all-cause mortality [5,10,11].

PWV is a gold standard measure in assessing arterial stiffness [12]. Carotid femoral PWV, a surrogate marker of aortic stiffness, is a well-established independent predictor of cardiovascular events and all-cause mortality in general and post-MI populations [13,14,15]. Local carotid PWV strongly correlates with carotid femoral PWV (r = 0.8) and may be potentially used as an easily-accessible alternative [16]. Increased aortic PWV shifts reflection waves’ return to the LV from diastole to systole, thus elevating LV afterload, oxygen demand, and stimulating LV hypertrophy [9]. Furthermore, earlier reflection waves’ return lowers diastolic pressure, thereby reducing coronary flow and posing a risk of myocardial ischemia.

The PWV/GLPSS index closely resembles the VA coupling ratio by combining the finest metrics of arterial and cardiac performance [9]. A study by Ikonomidis et al. showed that a novel index incorporating carotid femoral PWV and GLPSS outperforms the Ea/Ees ratio in predicting hypertension-mediated organ damage [17]. Moreover, local carotid PWV to GLPSS index was found as an independent predictor of cardiovascular adverse events in ACS survivors [4].

The reservoir–excess pressure paradigm offers an approximate global representation of arterial characteristics and their interconnections [18]. The model posits that the measured blood pressure is the sum of two components: the reservoir pressure and the excess pressure (Figure 1). Reservoir pressure reflects the behavior of compliant arteries that accumulate and release blood, while excess pressure correlates with blood flow and represents additional LV work [19]. Excess_PTI_ has been evidenced to predict target organ damage, adverse cardiovascular events, and/or all-cause mortality in post-ACS [6], heart failure [20,21], hypertensive [22,23], diabetic [24,25], and end-stage renal disease patients so far [26].

Our previous research demonstrated the potential of novel hemodynamic indices, such as VA coupling, the PWV/GLPSS index, and the excess_PTI_, to predict mortality risk in patients with ACS within three to five years post-event [4,5,6].

The present study aims to investigate whether these indices retain their predictive value over an extended ten-year follow-up period.

## 2. Materials and Methods

The study population consisted of 569 consecutive patients with ACS admitted to the hospital within 12 h of the onset of symptoms and with >50% stenosis of the coronary artery. Almost 41% of the participants were diagnosed with ST-elevation myocardial infarction (STEMI), while the remaining 59% suffered from non-STEMI or unstable angina. Nearly one-fourth of the individuals (23%, 133 patients) had a history of prior myocardial infarction that had been treated with percutaneous coronary intervention (PCI; 18%, 105 patients), coronary artery bypass grafting (CABG; 4%, 25 patients), or conservatively (<1%, 3 patients). The most prevalent comorbidities in the studied cohort were hypertension (80%), overweight or obesity (70%), and diabetes (32%). Additionally, nearly 41% of the participants declared current tobacco use. At baseline, the vast majority of patients received aspirin (98%), clopidogrel (98%), statins (97%), angiotensin-converting enzyme inhibitors/angiotensin receptor blockers (ACEI/ARB; 96%), and beta-blockers (92%). Furthermore, over a quarter of individuals used mineralocorticoid receptor antagonist (29%) and diuretic (26%), while 13% were administered long-acting nitrate. Patients were excluded from the study if they presented with cardiogenic shock, advanced and refractory heart failure, dialysis, or neoplasm. In addition, we decided to exclude individuals with current atrial flutter/fibrillation, as in many cases, a high heart rate and pulse wave variability precluded credible measurements of the hemodynamic indices studied. Patients with paroxysmal atrial flutter/fibrillation who presented with a sinus rhythm during hemodynamic measurements were included in our study. The primary endpoint of the current study was all-cause mortality only.

All echocardiographic and hemodynamic measurements were collected within 48 to 72 h after admission. One accredited expert in the field of echocardiography (A.M.) characterized with 3–6% intraobserver variability acquired all echocardiographic images and cine loops, and executed post-processing analyses. Moreover, A.M. performed all carotid ultrasound imaging. At the moment of inclusion, blood pressure was well-controlled with a median value of systolic and diastolic blood pressure of 113 mmHg (103–127 mmHg) and 68 mmHg (61–75 mmHg), respectively.

The study was performed with an ultrasound system (MyLab Class C, Esoate, Genova, Italy) equipped with a 3.0 MHz transducer. All imaging was performed with the subjects in the left lateral decubitus position. Both M-mode and two-dimensional (2D) echocardiographic images were captured from various projections, including the parasternal long-axis and short-axis views, as well as apical four-chamber, three-chamber, two-chamber, and five-chamber perspectives. The digital images collected were transferred to a computer workstation (MyLab Desk 5.0, Esaote, Genova, Italy) for subsequent offline analysis. Cardiac chamber dimensions, volumes, and wall thickness were measured per EACVI/ASE recommendations [27]. Left ventricular volumes and ejection fraction were calculated using the modified Simpson’s rule, averaging values obtained from both apical four-chamber and two-chamber views.

Cine loops synchronized with the electrocardiogram and optimized for speckle tracking were recorded from the apical four-chamber, three-chamber, and two-chamber views. To ensure optimal imaging of the left ventricular myocardium, settings such as gain, depth, and sector width were adjusted as needed. Speckle tracking analysis was carried out using specialized software (MyLab Desk 5.0, Esaote, Genova, Italy). The endocardial borders at the LV apex, basal segments, and the central point of the LV chamber were manually identified for each apical view, while the remaining myocardial segments were automatically delineated by the software, with manual corrections applied where necessary. The longitudinal strain at peak systole was determined for six myocardial segments per view, yielding a total of 18 measurements per patient. Global longitudinal peak systolic strain (GLPSS) was calculated by averaging the 18 segmental strain values from all three apical views.

To evaluate the end-systolic pressure-volume relationship (Figure 2), several parameters were measured, including LV end-systolic pressure, stroke volume, and key timing intervals (such as total ejection time, pre-ejection time, and the total duration of systole). These measurements were then utilized to calculate the LV end-systolic elastance (Ees), a measure of LV stiffness, using the single-beat approach as outlined by Chen et al. [28]. LV end-systolic pressure was derived by applying the formula 0.9 × systolic blood pressure. Brachial blood pressure was measured using an oscillometric device (Omron 705IT, Omron Healthcare Co., Ltd., Kyoto, Japan), while the subject was lying supine, after a resting period of 5 min. The value of Ees, representing the slope of the end-systolic pressure-volume curve, serves as an indicator of LV contractility. Effective arterial elastance (Ea) was calculated by dividing end-systolic pressure by stroke volume (SV). SV was determined by multiplying left ventricular outflow tract (LVOT) area and LVOT velocity-time integral (VTI). The Ea index is directly influenced by both heart rate and total peripheral resistance. Additionally, the ratio of Ea to Ees, known as the ventricular–arterial coupling ratio (Ea/Ees), was determined.

Radial pressure waveforms were captured noninvasively using a Piezoelectric tonometer (Colin BPM 7000, Colin Medical Instruments, Komaki, Japan). The analog signal was transmitted in real-time to a SphygmoCor Mx Aortic BP Monitoring System (AtCor Medical, Sydney, Australia), where it was processed using a validated transfer function to recreate a pressure waveform representative of the ascending aorta. Pulse-wave analysis was then conducted to evaluate both peripheral and central hemodynamics. Central pressure indices were derived using the proprietary software provided by AtCor Medical. If radial access was used during angiography, measurements were taken from the contralateral arm. All procedures were conducted by an experienced investigator (T.K.) and measurements were repeated until they met the software’s quality control standards. The resulting data were saved as text files and subsequently processed and analyzed using a custom Python program, based on published methodologies [19]. Specifically, we calculated the pressure-time integral for both the excess and reservoir pressure components (expressed in mmHg·ms; Figure 1) [6].

A high-resolution 4–13 MHz linear transducer (MyLab Class C, Esaote, Genova, Italy) was used to perform ultrasound imaging of the left common carotid artery in all participants. The measurements were taken 1 cm before the bifurcation of the left common carotid artery and analyzed using ArtLab software (MyLab Desk 5.0, Esaote, Genova, Italy), which utilizes radiofrequency data technology (QAS^RF^) to assess arterial stiffness. The ArtLab system cycles through six consecutive measurements of both the arterial diameter and distensibility, maintaining a standard deviation of less than 35 μm for each cardiac cycle. The results were averaged over each cycle to obtain consistent data.

Arterial wall stiffness was quantified by calculating the local carotid pulse wave velocity (PWV), a key marker of arterial compliance. A risk index was estimated as PWV divided by GLPSS [4]. PWV was computed using the following formula:PWV = 1/√(ρ × DC) = √((D^2^ × Δp)/(ρ(2D × ΔD + ΔD^2^)))
where D represents the diastolic diameter, ΔD is the change in diameter during systole, DC is the distensibility coefficient, Δp is the local pulse pressure, and ρ is the blood density.

Continuous data are presented as mean ± standard deviation (SD). We estimated survival curves using the Kaplan–Meier method and compared them with the log-rank method. The predefined primary outcome was death from all causes. We constructed tertiles from the entire dataset of hemodynamic markers in the studied population. To identify the outcome variable, we applied the Cox proportional hazard method (SPSS version 29, IBM Corp, Armonk, NY, USA). All tests were two-tailed, and *p*-values < 0.05 were considered significant.

## 3. Results

Of the 569 subjects included (Table 1), the mean age ± SD was 63.4 ± 10.7, 403 were male (71%), and 166 were female (29%). At the examination, 133 had a history of myocardial infarction, 25 coronary artery bypass grafting (CABG), 454 suffered from hypertension, and 184 had diabetes. During a median follow-up of 3249 days, 172 subjects reached the endpoint event (death).

The deceased patients (Table 2) were older (69.5 ± 10.5 years vs. 60.8 ± 9.7 years, *p* < 0.001), had lower ejection fraction (45.7 ± 13.1% vs. 53.0 ± 10.3%, *p* < 0.001), impaired GLPSS −12.9 ± 4.4% vs. −15.1 ± 3.6%, *p* < 0.001), higher VA coupling (1.9 ± 0.9 vs. 1.5 ± 0.4, *p* < 0.001), higher excess_PTI_ (4259.3 ± 1498.6 mmHg·ms vs. 3906.1 ± 1172.5 mmHg·ms, *p* < 0.001), and lower PWV/GLPSS index (−0.92 ± 0.52 m/s % vs. −0.66 ± 0.26 m/s %, *p* < 0.001).

Each of the above metrics was categorized according to the tertile of its distribution. Kaplan–Meier plot analysis demonstrates that the last tertile of EF (<47%), GLPSS (>−12.8%), VA coupling (>1.9), excess_PTI_ (>4382 mmHg·ms), and PWV/GLPSS index (<−0.77 m/s %) were associated with increased risk of death (*p* < 0.01 in the log-rank test, Figure 3). In Table 3, we present the Cox proportional hazards model results for the association between hemodynamic metrics and risk of death (all-cause mortality). In this model, ejection fraction, VA-coupling, excess_PTI_, and PWV/GLPSS index were associated independently with all-cause mortality during decennial follow-up.

In a subsequent analysis using Cox models, we assessed critical clinical variables. The results indicated that a history of myocardial infarction (MI) was significantly associated with an increased risk, with a hazard ratio (HR) of 2.1 (95% CI: 1.6–2.9). Similarly, age demonstrated a substantial association, with an HR of 1.06 (95% CI: 1.05–1.08). In contrast, neither stroke nor diabetes showed a significant relationship with increased mortality.

Given that ageing may affect the hemodynamic parameters under investigation, we constructed an additional multivariate regression model that included age as a variable. In this model, age, along with the last tertiles of EF, VA coupling, and excess_PTI_, emerged as independent predictors of all-cause mortality (Table 4). Notably, the PWV/GLPSS ratio appeared statistically insignificant after incorporating age in the analysis.

## 4. Discussion

This study demonstrates that specific hemodynamic indices—VA coupling, the pulse wave velocity-to-global longitudinal strain ratio, and the excess pressure-time integral—are significant predictors of all-cause mortality in long-term survivors of acute coronary syndrome. While the widespread use of reperfusion therapies like percutaneous coronary intervention, combined with pharmacotherapy, has reduced mortality in ACS, the population remains at high risk due to advanced age and comorbidities, such as hypertension, diabetes, and chronic kidney disease. Recent research by Spadafora et al. revealed a higher prevalence of comorbidities and older age in non-STEMI, but comparable clinical outcomes in both STEMI and non-STEMI groups one year after myocardial infarction [29].

Over a ten-year follow-up, 30% of our patients died, consistent with previous findings. A similar study on 100,601 patients who survived myocardial infarction (MI) between 2005 and 2012 reported 31,622 all-cause deaths and 12,901 cardiovascular deaths over a mean follow-up of 4.0 years [1]. In both studies, advanced age was a consistent predictor of all-cause mortality. In our cohort, deceased patients were significantly older and had worse hemodynamic profiles, including lower ejection fraction, unfavorable VA coupling, higher excess_PTI_, and a reduced PWV/GLPSS index than survivors.

Previous research on this population highlighted the link between hemodynamic indices and adverse outcomes. In one study with a follow-up of 625 days, elevated VA coupling was strongly associated with an increased risk of death, stroke, and recurrent myocardial infarction [5]. In a subsequent investigation conducted over 1316 days, the PWV/GLPSS ratio emerged as a significant predictor of adverse events [4]. Correspondingly, another analysis of 251 ACS survivors (median follow-up of 1245 days) found that excess_PTI_ was a key predictor of poor outcomes, including death, stroke, and myocardial infarction [6]. In the current study, with a median follow-up of 3249 days, 172 patients experienced the endpoint event of all-cause death. Unlike previous studies, this research focused solely on mortality. The deceased patients had a more advanced age and worse hemodynamic profiles. Kaplan–Meier analysis showed that the lowest tertile of EF, unfavorable VA coupling, elevated excess pressure, and a lower PWV/GLPSS ratio were significantly associated with an increased risk of death. Multiple Cox regression model confirmed these metrics as independent predictors of mortality, although PWV/GLPSS ratio did not reach statistical significance when including age in the multivariate analysis.

The results of our study align with a recently published paper showing that increased VA coupling was independently associated with higher in-hospital and one-year mortality in 4685 patients admitted to a cardiac intensive care unit, predominantly due to ACS and/or heart failure [30]. VA coupling exceeding 2.0 predicted the worst outcomes (HR 1.55; 1.31–1.82). Notably, measurement of left-ventricular end-systolic volume was simplified to the Teichholz formula. Similarly, research by Trambaiolo et al. demonstrated that elevated VA coupling was linked with higher early and one-year mortality in 48 STEMI individuals [11]. Furthermore, raised VA coupling levels were predictive of norepinephrine use after PCI. Conversely, VA coupling significantly improved after administration of levosimendan, particularly through greater reduction in Ea than Ees. In addition, the results of one-year observation in 84 post-STEMI patients disclosed that lifted VA coupling surpassed EF and GLPSS in major adverse cardiac events (MACE) prediction and remained an independent prognostic factor in multivariate analysis [10]. Unfavorable outcomes were connected with VA coupling values of 1.71 and higher. Moreover, VA coupling was reported to be an independent correlate of B-type natriuretic peptide (BNP) and related to cardiovascular mortality during a five-year follow-up period in 41 post-ACS individuals [31]. However, VA coupling did not exceed the predictive value of BNP. Additionally, VA coupling assessment by cardiac magnetic resonance demonstrated a significant association of VA coupling levels with the incidence of MACE in 1235 ACS survivors [32]. Besides, VA coupling correlated well with infarct size and GLPSS. On the other hand, VA coupling was not associated with MACE in the RIGID-MI cohort of 374 patients with EF ≥ 40%, measured one month after MI [33]. Only the PWV/GLPSS ratio appeared as an independent MACE determinant in this population within 32 months of follow-up, albeit missing data on PWV/GLPSS concerned over 30% of the cohort. Unexpectedly, exercise-based cardiac rehabilitation did not alter VA coupling ratio, although significant improvements in EF and GLPSS were noted [34].

Interestingly, we did not find a significant relationship between GLPSS and all-cause mortality, although similar evidence is reported in the literature. Two recent publications proved that composite endpoints including death and heart failure rehospitalization, as well as all-cause mortality and new ACS occurrence, were associated with both EF and GLPSS in ACS survivors during long-term follow-up (941 and 1436 patients, respectively) [35,36]. However, GLPSS did not add predictive value to EF, even in a preserved EF population.

Our study has several limitations, starting from a relatively small sample size observed in a single center and a high male-to-female ratio, which may disturb extrapolation of the results to the female population. Furthermore, we used vendor-specific software for GLPSS and local carotid PWV calculation, which may affect the comparability of the findings across different software suppliers. The primary outcome of our study was all-cause mortality; however, also assessing cardiovascular events and cardiovascular deaths would be beneficial. Moreover, we could not collect reliable data on EF as well as blood pressure and glycemia control before inclusion in the study. Besides, we did not analyze changes in pharmacotherapy during the follow-up period. Eventually, the proportional hazards assumption was not reported.

## 5. Conclusions

In conclusion, this 10-year study highlights the persistent risk of mortality and adverse cardiovascular outcomes in ACS survivors, especially in older individuals with poor hemodynamic parameters. These findings emphasize the importance of long-term monitoring and development of tailored treatment strategies to mitigate cardiovascular risk in this vulnerable population. Hemodynamic markers, such as VA coupling, excess_PTI_, and PWV/GLPSS should be acknowledged as independent predictors of mortality in post-ACS patients. Further research is needed to explore their clinical utility, including risk stratification and guiding therapeutic decisions.

## Figures and Tables

**Figure 1 jcm-14-06627-f001:**
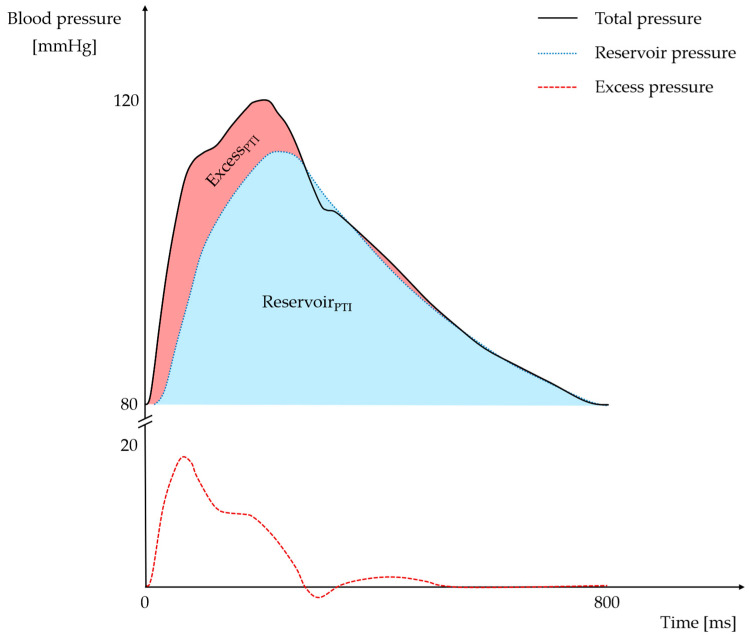
The excess-reservoir pressure model assumes that the measured total blood pressure is a sum of excess and reservoir pressures. Appropriate areas under the curves may be used for calculating excess and reservoir pressure-time integrals. Excess_PTI_: excess pressure-time integral; Reservoir_PTI_: reservoir pressure-time integral.

**Figure 2 jcm-14-06627-f002:**
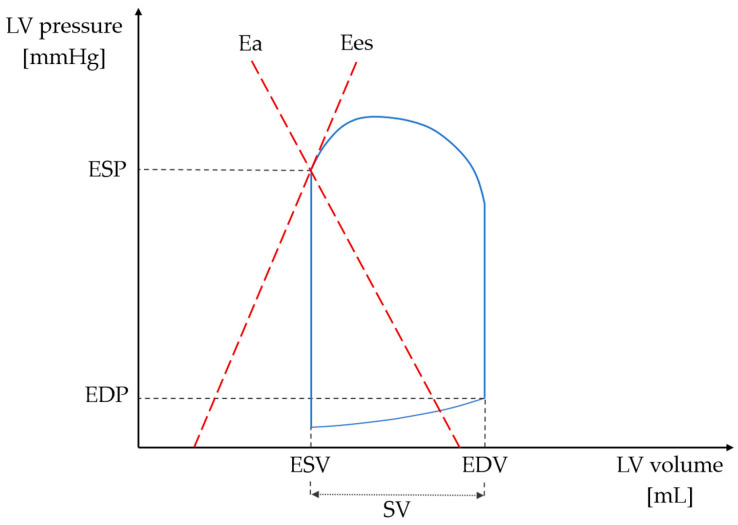
In-depth analysis of left ventricular pressure-volume loop characteristics enables the assessment of the ventricular–arterial coupling ratio (Ea/Ees). Ea is computed as ESP/SV, while ESP/ESV equals Ees. Ea: effective arterial elastance; EDP: end-diastolic pressure; EDV: end-diastolic volume; Ees: end-systolic elastance; ESP: end-systolic pressure; ESV: end-systolic volume; LV: left-ventricular; SV: stroke volume.

**Figure 3 jcm-14-06627-f003:**
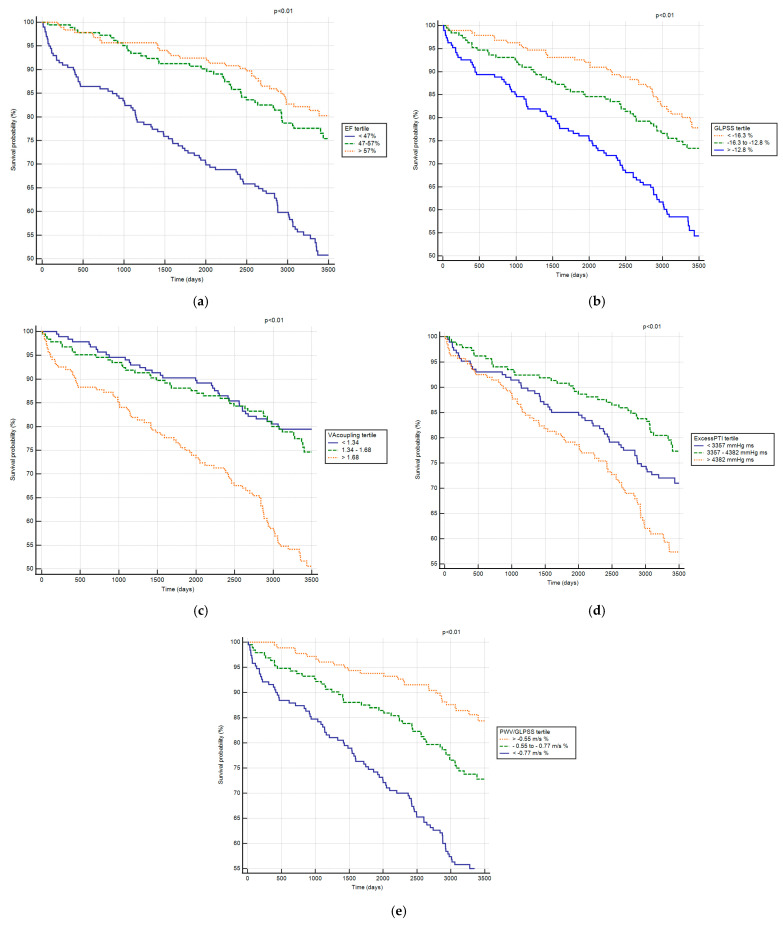
Kaplan–Meier survival estimates of all-cause mortality according to tertiles of echocardiographic metrics: (**a**) ejection fraction (EF) with tertile cut-points: 47–57%; (**b**) global longitudinal peak systolic strain (GLPSS) with tertile cut-points: −16.3 to −12.8%; (**c**) ventricular–arterial coupling (VA coupling) with tertile cut-points: 1.34–1.68; aortic pressure analysis: (**d**) excess pressure-time integral (excess_PTI_) with tertile cut-points: 3357–4382 mmHg·ms; and (**e**) pulse wave velocity/global longitudinal peak systolic strain (PWV/GLPSS) index with tertile cut-points: −0.77 to −0.55 m/s %. All comparisons were significant, with *p* < 0.01 using a log-rank test.

**Table 1 jcm-14-06627-t001:** Clinical characteristics of the studied population.

Characteristic	All Subjects	Survivors	Deceased
Male/Female	403/166	280/116	122/50
Age (years ± SD)	63.4 ± 10.7	60.8 ± 9.7	69.5 ± 10.5
MI in the past	133	68	65
CABG history	25	12	13
Hypertension	454	307	147
Diabetes	184	117	67

CABG: coronary artery bypass grafting; MI: myocardial infarction; SD: standard deviation.

**Table 2 jcm-14-06627-t002:** Hemodynamic characteristics of the studied population.

Parameter	Survivors	Deceased	*p*-Value
EF (%)	53.0 ± 10.3	45.7 ± 13.1	<0.001
GLPSS (%)	−15.1 ± 3.6	−12.9 ± 4.4	<0.001
VA coupling	1.5 ± 0.4	1.9 ± 0.9	<0.001
Excess_PTI_ (mmHg·ms)	3906.1 ± 1172.5	4259.3 ± 1498.6	<0.001
PWV/GLPSS (m/s %)	−0.66 ± 0.26	−0.92 ± 0.52	<0.001

EF: ejection fraction; Excess_PTI_: excess pressure-time integral; GLPSS: global longitudinal peak systolic strain; PWV: pulse wave velocity; VA coupling: ventricular–arterial coupling.

**Table 3 jcm-14-06627-t003:** Results of Cox proportional hazards regression model for the association between tested metrics and risk of the endpoint event (all-cause mortality).

Parameter	Multivariate	
	HR (95% CI)	*p*-Value
GLPSS > −12.8%	0.97 (0.84–1.11)	0.640
PWV/GLPSS < −0.77 m/s %	1.77 (1.16–2.71)	0.008
Excess_PTI_ > 4382 mmHg·ms	1.27 (1.14–1.41)	0.001
VA coupling > 1.9	1.17 (1.03–1.33)	0.020
EF < 47%	1.94 (1.32–2.87)	0.001

CI: confidence intervals; EF: ejection fraction; Excess_PTI_: excess pressure-time integral; GLPSS: global longitudinal peak systolic strain; PWV: pulse wave velocity; VA coupling: ventricular–arterial coupling.

**Table 4 jcm-14-06627-t004:** Results of Cox proportional hazards regression model for the association between tested metrics, including age, and risk of the endpoint event (all-cause mortality).

Parameter	Multivariate	
	HR (95% CI)	*p*-Value
GLPSS > −12.8%	0.99 (0.87–1.14)	0.959
PWV/GLPSS < −0.77 m/s %	1.31 (0.86–1.99)	0.203
Excess_PTI_ > 4382 mmHg·ms	1.18 (1.06–1.32)	0.003
VA coupling > 1.9	1.16 (1.02–1.31)	0.020
EF < 47%	1.92 (1.31–2.82)	0.001
Age	1.05 (1.04–1.07)	0.001

CI: confidence intervals; EF: ejection fraction; Excess_PTI_: excess pressure-time integral; GLPSS: global longitudinal peak systolic strain; PWV: pulse wave velocity; VA coupling: ventricular–arterial coupling.

## Data Availability

The data presented in this study are available from the corresponding author on reasonable request.

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
