# Peer review of "Hemodynamic Markers Predict Outcomes a Decade After Acute Coronary Syndrome"

_jcm, 2025, doi:10.3390/jcm14186627_

Round 1

Reviewer 1 Report

Comments and Suggestions for Authors

Introduction:

Author should clearly state how the non invasive markers are taken in real world. For example, what are steps to take VA coupling, PWV/GLS ration? also what are implication of such as they are usually not done in real world. There are a lot of other markers done such as for ACS there are a lot of invasive hemodynamic records. need to explain why shall we choose such markers which are rarely done.   

Materials and method: Primary outcome was death from all causes”. It would be better presentation if death from cardiovascular disease were the outcome as markers are indicators of cardiovascular changes which finally leads to adverse outcome like death.  Death from other cuases may under estimate or overestimate the prediction of markers taken at the begninning of the study.   

Result: Male to female ratio is not uniform which may cause under representation of female and result may not be extrapolated to female population. previous hx of MI may need to low EF in that case prediction of long term mortality with predictors will be difficult due to unknown baseline markers. It needs to clarify what was the EF before the index presentation of ACS. Also HTN is an important factor which can alter the markers, particulary uncontrolled HTN. 454 patients had HTN but it is important to know how many of them has uncontrolled HTN or uncontrolled DM as they are independent predictors for long term mortality and may confound the result.   

DIscussion: "more advanced age and worse hemodynamic profiles. Kaplan-Meier analysis showed that the lowest tertile of EF, unfa- vorable VA coupling, elevated excess pressure, and a lower PWV/GLPSS ratio”. advance age, worse hemodynamic profile patients will have low EF and other predictors used in this study as they are interdependent. 

Comments on the Quality of English Language

Good to excellent

Author Response

Dear Reviewer,

the authors would like to thank you for your valuable comments, which helped to improve the quality of the manuscript.

Please find our detailed revision note below.

Sincerely yours,

Authors

Reviewer 1:

“Introduction: Author should clearly state how the non invasive markers are taken in real world. For example, what are steps to take VA coupling, PWV/GLS ration? also what are implication of such as they are usually not done in real world. There are a lot of other markers done such as for ACS there are a lot of invasive hemodynamic records. need to explain why shall we choose such markers which are rarely done.”

Answer: The uppermost advantage of the studied parameters is the fact that all of them are measured non-invasively, which makes them safe, easy-accessible, and cost-effective. This fact has been emphasized in the Introduction section of the revised manuscript (page 2, lines 47–49). Currently, they are not routinely measured in real-world settings since no studies have explored their clinical utility, including risk stratification and guiding therapeutic decisions, although a few studies showed their prognostic potential in relatively small post-ACS populations. In the world of evidence-based medicine, large, multicenter studies are needed to introduce these parameters to the guidelines and subsequently to wide clinical use. We believe that our results, combining different novel hemodynamic markers simultaneously, could serve as a background and inspiration for further research.

All studied parameters can be measured by incorporating echocardiography, carotid ultrasound, tonometry, and blood pressure values. Echocardiographic assessment requires measurements of EF, GLPSS, LVOT area, LVOT VTI, and key timing intervals, such as total ejection time, pre-ejection time, and the total duration of systole. Carotid ultrasound using ArtLAB software is enough to assess local carotid PWV. The PWV/GLPSS ratio is a simple mathematical equation. ExcessPTI calculation requires radial artery tonometry and post-processing analysis. So as not to duplicate, precise measurement techniques are presented in the Materials and Methods section (pages 3–6, lines 92–199).

“Materials and method: Primary outcome was death from all causes”. It would be better presentation if death from cardiovascular disease were the outcome as markers are indicators of cardiovascular changes which finally leads to adverse outcome like death.  Death from other cuases may under estimate or overestimate the prediction of markers taken at the begninning of the study.”

Answer: During the prolonged follow-up period, a significant number of deaths occurred at home. In many cases, there were no health check-ups directly before death and no actual medical records. Furthermore, Poland is a country with a high percentage of garbage codes used in public death statistics, especially cardiovascular ones [1]. Taken together, we found it impossible to precisely classify the causes of death. Therefore, we decided to focus on all-cause mortality only in order to maintain the reliability of the data and sample size. 

[1] Fihel A., Muszyńska-Spielauer M.M. Using multiple cause of death information to eliminate garbage codes. Demographic Research, 2021, 45, 345-360.

“Result: Male to female ratio is not uniform which may cause under representation of female and result may not be extrapolated to female population. “

Answer: We included 569 consecutive ACS patients to minimize selection bias, provide transparency, and ensure reproducibility. The male-to-female ratio in our study (71%/29%) reflects real-world practice and aligns with large population studies, such as the UK Biobank study, which enrolled almost 472,000 participants and observed a male-to-female ratio of 71.2% to 28.8% in MI incidence [2]. Nevertheless, we agree that the number of males and females included in our study differs significantly and may disturb extrapolation of results to the female population. This fact has been emphasized in the last paragraph of the Discussion section regarding our study limitations (page 10, lines 311–320). We will consider the additional analyses in the female subpopulation in our future work.

[2] Millett E.R.C., Peters S.A.E., Woodward M. Sex differences in risk factors for myocardial infarction: cohort study of UK Biobank participants. BMJ, 2018, 363, k4247.

“previous hx of MI may need to low EF in that case prediction of long term mortality with predictors will be difficult due to unknown baseline markers. It needs to clarify what was the EF before the index presentation of ACS.”

Answer: Among 569 participants included in our study, 133 individuals had a history of myocardial infarction in the past. The majority of them were treated for prior MI in other hospitals and presented to our department for the first time during the next event. Therefore, we did not have their echocardiographic images stored in our archives. Moreover, a substantial part of the patients did not provide us with their echocardiographic documentation. Furthermore, there was a group of patients who did not have an echocardiography done within months before hospitalization in our department. Some patients delivered discharge cards from prior ACS hospitalization with echocardiographic imaging included; however, these results may have changed significantly from the date of discharge due to cardiac remodeling, the influence of revascularization, rehabilitation, and pharmacotherapy applied. Taken together, we could not collect reliable data on EF before the presentation with MI in our department. Nevertheless, we agree that historical baseline EF could affect our results, which has been underscored in the last paragraph of the Discussion section regarding limitations (page 10, lines 311–320). We plan to focus solely on the first MI subpopulation in our future publication.

“Also HTN is an important factor which can alter the markers, particulary uncontrolled HTN. 454 patients had HTN but it is important to know how many of them has uncontrolled HTN or uncontrolled DM as they are independent predictors for long term mortality and may confound the result.”

Answer: At the moment of inclusion (within 48 to 72 hours after admission), blood pressure was well-controlled with a median value of systolic and diastolic blood pressure of 113 mmHg (103-127 mmHg) and 68 mmHg (61-75 mmHg), respectively. The majority of patients did not provide us with their home blood pressure monitoring records. Although most of them claimed that their home blood pressure was within the recommended values, we could not take it as a certainty. If the patient presented with elevated blood pressure, we introduced or intensified pharmacotherapy from the first day of hospitalization, which may explain the high participation of well-controlled hypertensive patients. However, it should be noted that blood pressure control could change after discharge or during a decennial follow-up. The information on the baseline blood pressure has been added to the Materials and Methods section (page 4, lines 118–120).

With regard to DM control, routine measurement of HbA1c when recruiting study participants was unavailable in our institution due to economic reasons (our study received no external funding). The majority of patients did not provide us with their home glucose monitoring records. Although most of them claimed that their home glucose was within the recommended values, we could not take it as a certainty. If the patient presented with elevated glucose levels, we introduced or intensified pharmacotherapy from the first day of hospitalization. In a few cases of uncontrolled diabetes despite our treatment modifications, we transferred patients to the Diabetes Department for further glycemia management. Thus, we could assume that glycemia was well-controlled at least at the beginning of our observation. However, it should be noted that glycemia control could change after discharge or during a decennial follow-up.

“DIscussion: "more advanced age and worse hemodynamic profiles. Kaplan-Meier analysis showed that the lowest tertile of EF, unfa- vorable VA coupling, elevated excess pressure, and a lower PWV/GLPSS ratio”. advance age, worse hemodynamic profile patients will have low EF and other predictors used in this study as they are interdependent.”

Answer: The term “hemodynamic profile” used in the Discussion section refers to the studied hemodynamic parameters, such as EF, VA coupling, excess pressure, and PWV/GLPSS ratio. When using this term, we did not introduce any new or unspecified hemodynamic parameter different than the studied ones. Thus, “worse hemodynamic profile” refers to and combines a lower EF, unfavorable (higher) VA coupling, elevated excess pressure, and a lower PWV/GLPSS ratio. These parameters could not be recognized as interdependent, as they were found to be the independent predictors in the multivariate regression model (Table 3, p<0.05).

With regard to age, we constructed an additional multivariate regression model that included age as a variable. In this model, age, along with the last tertiles of EF, VA coupling, and excessPTI, emerged as independent predictors of all-cause mortality (p<0.05). Notably, the PWV/GLPSS ratio appeared statistically insignificant after incorporating age in the analysis. The results of the additional analysis have been added to the Results section and are summarized in Table 4 (page 8, lines 236–245).

Reviewer 2 Report

Comments and Suggestions for Authors

This prospective observational study enrolled 569 consecutive survivors of acute coronary syndrome and followed them for a median of ~9 years. At baseline the investigators measured LVEF, GLPSS, VA coupling, excessPTI, and the ratio of carotid pulse wave velocity to global longitudinal strain. During follow‑up, 172 participants died. Deceased subjects were older and had lower LVEF, worse GLPSS, higher VA coupling, higher excessPTI, and lower PWV/GLPSS. Kaplan–Meier curves showed that the worst tertile of each marker was associated with increased mortality. In a multivariable Cox model including these hemodynamic indices, LVEF < 47 %, VA coupling > 1.9, excessPTI > 4382 mmHg·ms and PWV/GLPSS < –0.77 m/s % were independently associated with all‑cause mortality. GLPSS alone did not remain significant after adjustment. The authors conclude that these novel markers, especially when combined, may improve long‑term risk stratification in ACS survivors.

A median follow‑up of 8–9 years and 172 events provide reasonable power to detect associations. The investigators combined echocardiographic assessment, tonometric pressure waveform analysis and carotid ultrasound, allowing exploration of interrelated cardiac and arterial factors. Few prior studies have assessed ventricular–arterial coupling, excess pressure‑time integral and the PWV/GLPSS index simultaneously in a post‑ACS cohort.

Dichotomising continuous variables can diminish prognostic information; however, the authors appropriately explored risk across tertiles with Kaplan–Meier curves and then included the continuous markers in Cox models. The hazard ratios reported for LVEF, VA coupling, excessPTI and PWV/GLPSS remained significant after adjustment, suggesting independent associations rather than mere collinearity.

Major concerns:

The hemodynamic measurements were performed within days after ACS, but the precise timing relative to the index event and revascularization is not stated. 

Use of guideline‑directed medical therapy strongly influences long‑term outcomes. The manuscript does not report baseline or discharge medications, nor adjust for them in the multivariable models.

The discussion implies that VA coupling, excessPTI and PWV/GLPSS may be used to tailor therapy, however the study is observational. Whether modifying these parameters improves outcomes is unknown.

The authors cite several of their own prior studies but there are other studies to consider. For example, a 2024 prospective study of 1680 ACS patients found that GLS and LVEF were both independent predictors of mortality, but GLS did not significantly improve risk prediction over LVEF. Meanwhile, a 2024 three‑dimensional echocardiographic study reported that ventricular‑arterial coupling > 1.71 strongly predicted adverse events in young STEMI patients

The assertion that these indices may open new possibilities for personalized medicine is a bit speculative. Before incorporating PWV/GLPSS or excessPTI into routine care, validation in independent, multicenter cohorts and demonstration of clinical utility are required.

No data are provided on intra‑ and interobserver variability for echocardiographic strain or PWV measurements.

Author Response

Dear Reviewer,

the authors would like to thank you for your valuable comments, which helped to improve the quality of the manuscript.

Please find our detailed revision note below.

Sincerely yours,

Authors

Reviewer 2:

This prospective observational study enrolled 569 consecutive survivors of acute coronary syndrome and followed them for a median of ~9 years. At baseline the investigators measured LVEF, GLPSS, VA coupling, excessPTI, and the ratio of carotid pulse wave velocity to global longitudinal strain. During follow‑up, 172 participants died. Deceased subjects were older and had lower LVEF, worse GLPSS, higher VA coupling, higher excessPTI, and lower PWV/GLPSS. Kaplan–Meier curves showed that the worst tertile of each marker was associated with increased mortality. In a multivariable Cox model including these hemodynamic indices, LVEF < 47 %, VA coupling > 1.9, excessPTI > 4382 mmHg·ms and PWV/GLPSS < –0.77 m/s % were independently associated with all‑cause mortality. GLPSS alone did not remain significant after adjustment. The authors conclude that these novel markers, especially when combined, may improve long‑term risk stratification in ACS survivors.

A median follow‑up of 8–9 years and 172 events provide reasonable power to detect associations. The investigators combined echocardiographic assessment, tonometric pressure waveform analysis and carotid ultrasound, allowing exploration of interrelated cardiac and arterial factors. Few prior studies have assessed ventricular–arterial coupling, excess pressure‑time integral and the PWV/GLPSS index simultaneously in a post‑ACS cohort.

Dichotomising continuous variables can diminish prognostic information; however, the authors appropriately explored risk across tertiles with Kaplan–Meier curves and then included the continuous markers in Cox models. The hazard ratios reported for LVEF, VA coupling, excessPTI and PWV/GLPSS remained significant after adjustment, suggesting independent associations rather than mere collinearity.

Major concerns:

“The hemodynamic measurements were performed within days after ACS, but the precise timing relative to the index event and revascularization is not stated.”

Answer: All echocardiographic and hemodynamic measurements were collected within 48 to 72 hours after admission. This information has also been added to the Materials and Methods section (page 4, lines 114–115).

“Use of guideline‑directed medical therapy strongly influences long‑term outcomes. The manuscript does not report baseline or discharge medications, nor adjust for them in the multivariable models.”

Answer: The use of the relevant medications at baseline has already been presented in the original version of the manuscript (page 3, lines 100-102):

“At baseline, the vast majority of patients received aspirin (98%), clopidogrel (98%), statins (97%), angiotensin-converting enzyme inhibitors/angiotensin receptor blockers (ACEI/ARB; 96%), and beta-blockers (92%).”

In the revised version of the manuscript, we have also added information on the following drug groups (page 3, lines 104–106):

“Furthermore, over a quarter of individuals used mineralocorticoid receptor antagonist (29%) and diuretic (26%), while 13% were administered long-acting nitrate.”

Please note that SGLT2i and sacubitril/valsartan were approved for clinical use in 2014 and 2015, respectively, and were unavailable on the market at the moment of inclusion.

The list of medications used could also change after discharge or during long-term follow-up. Thus, we decided not to include medications in regression models.

“The discussion implies that VA coupling, excessPTI and PWV/GLPSS may be used to tailor therapy, however the study is observational. Whether modifying these parameters improves outcomes is unknown.”

Answer: We fully agree with your comment. The Conclusions section has been rewritten, according to your suggestions (page 10, lines 325–329).

“The authors cite several of their own prior studies but there are other studies to consider. For example, a 2024 prospective study of 1680 ACS patients found that GLS and LVEF were both independent predictors of mortality, but GLS did not significantly improve risk prediction over LVEF. Meanwhile, a 2024 three‑dimensional echocardiographic study reported that ventricular‑arterial coupling > 1.71 strongly predicted adverse events in young STEMI patients”

Answer: The discussion section has been substantially broadened with relevant evidence from other studies, including the suggested ones (pages 8–10, lines 253–255 and 276–310).

“The assertion that these indices may open new possibilities for personalized medicine is a bit speculative. Before incorporating PWV/GLPSS or excessPTI into routine care, validation in independent, multicenter cohorts and demonstration of clinical utility are required.”

Answer: We fully agree with your comment. The Conclusions section has been rewritten, according to your suggestions (page 10, lines 325–329).

“No data are provided on intra‑ and interobserver variability for echocardiographic strain or PWV measurements.”

Answer: One accredited expert in the field of echocardiography (A.M.) characterized with 3–6% intraobserver variability acquired all echocardiographic images and cine loops, and executed post-processing analyses. The assessment of intraobserver variability included GLPSS measurements. Moreover, A.M. performed all carotid ultrasound imaging.

Data on intraobserver variability has been added to the Materials and Methods section (page 4, lines 115–118).

Reviewer 3 Report

Comments and Suggestions for Authors

My congratulations to the authors because the manuscript deals with a clinically relevant question: whether advanced hemodynamic markers such as VA coupling, PWV/GLPSS and  excessPTI mantain their prognostic value in ACS patients over a 10-year follow-up.

The decision to exclude atrial fibrillation warrants further explanation, since AF is a frequent comorbidity in ACS and its omission may reduce the generalizability of the findings.

The reliance on vendor-specific software  should also be acknowledged as a limitation, as strain values may vary across different imaging platforms and are not universally interchangeable.

The manuscript indicates that tertiles of hemodynamic indices were used, but it is unclear whether these thresholds were derived from the study population itself or predefined based on previous literature. This point requires clarification by the authors.

The proportional hazards assumption for the Cox models was not reported. Given the extended 10-year follow-up, this verification is essential and must be explicitly addressed.

The discussion would benefit from elaboration on the clinical applicability of VA coupling and PWV/GLPSS in the contemporary management of ACS. Specifically, the authors should elaborate on whether these markers could enhance risk stratification beyond established tools such as the GRACE or TIMI scores. Moreover authors are encouraged to include that latest evidence of ACS prognosis in their discussion in addition to their findings (https://doi.org/10.1007/s40256-025-00739-8)

Author Response

Dear Reviewer,

the authors would like to thank you for your valuable comments, which helped to improve the quality of the manuscript.

Please find our detailed revision note below.

Sincerely yours,

Authors

Reviewer 3:

My congratulations to the authors because the manuscript deals with a clinically relevant question: whether advanced hemodynamic markers such as VA coupling, PWV/GLPSS and excessPTI mantain their prognostic value in ACS patients over a 10-year follow-up.

“The decision to exclude atrial fibrillation warrants further explanation, since AF is a frequent comorbidity in ACS and its omission may reduce the generalizability of the findings.”

Answer: We decided to exclude individuals with current atrial flutter/fibrillation, as in many cases, a high heart rate and pulse wave variability precluded credible measurements of the hemodynamic indices studied. Patients with paroxysmal atrial flutter/fibrillation who presented with a sinus rhythm during hemodynamic measurements were included in our study. The justification for current atrial flutter/fibrillation exclusion has been added to the Materials and Methods section (page 3, lines 108–112).

“The reliance on vendor-specific software should also be acknowledged as a limitation, as strain values may vary across different imaging platforms and are not universally interchangeable.”

Answer: We fully agree with your comment. A paragraph describing limitations has been added at the end of the Discussion section (page 10, lines 311–320).

“The manuscript indicates that tertiles of hemodynamic indices were used, but it is unclear whether these thresholds were derived from the study population itself or predefined based on previous literature. This point requires clarification by the authors.”

Answer: The tertile thresholds for all hemodynamic parameters were derived from the studied population itself. This fact has been emphasized in the revised version of the manuscript for clarity (page 6, line 197).

“The proportional hazards assumption for the Cox models was not reported. Given the extended 10-year follow-up, this verification is essential and must be explicitly addressed.”

Answer: We agree that the value of our results would improve by reporting the proportional hazard assumption. Therefore, we plan to perform proportional hazard assumption analysis using Schönfeld residuals. However, at the time of revisions, our statistician was on holiday and we were unable to perform the analysis. Hence, we asked the Editorial Office for a prolonged time for revisions so that we could comprehensively address your comment with the statistics done. Unfortunately, the Editorial Office did not agree to our suggested revision date. The lack of the proportional hazards assumption report is currently acknowledged as a limitation of our study (page 10, lines 311–320).

“The discussion would benefit from elaboration on the clinical applicability of VA coupling and PWV/GLPSS in the contemporary management of ACS. Specifically, the authors should elaborate on whether these markers could enhance risk stratification beyond established tools such as the GRACE or TIMI scores. Moreover authors are encouraged to include that latest evidence of ACS prognosis in their discussion in addition to their findings (https://doi.org/10.1007/s40256-025-00739-8)”

Answer: The discussion has been supplemented with evidence on the clinical applicability and prognostic value of VA coupling and PWV/GLPSS (pages 9–10, lines 278–310). In the addition, the latest evidence on ACS prognosis was included as suggested (page 8, lines 253–255).

Round 2

Reviewer 3 Report

Comments and Suggestions for Authors

Congratulations to the authors for having modified their papers according to my comments appropriately.